# OLMA: ONE LOSS FOR MORE ACCURATE TIME SERIES FORECASTING

## ABSTRACT

Time series forecasting faces two important but often overlooked challenges. Firstly, the inherent random noise in the time series labels sets a theoretical lower bound for the forecasting error, which is positively correlated with the entropy of the labels. Secondly, neural networks exhibit a frequency bias when modeling the state-space of time series, that is, the model performs well in learning certain frequency bands but poorly in others, thus restricting the overall forecasting performance. To address the first challenge, we prove a theorem that there exists a unitary transformation that can reduce the marginal entropy of multiple correlated Gaussian processes, thereby providing guidance for reducing the lower bound of forecasting error. Furthermore, experiments confirm that Discrete Fourier Transform (DFT) can reduce the entropy in the majority of scenarios. Correspondingly, to alleviate the frequency bias, we jointly introduce supervision in the frequency domain along the temporal dimension through DFT and Discrete Wavelet Transform (DWT). This supervision-side strategy is highly general and can be seamlessly integrated into any supervised learning method. Moreover, we propose a novel loss function named OLMA, which utilizes the frequency domain transformation across both channel and temporal dimensions to enhance forecasting. Finally, the experimental results on multiple datasets demonstrate the effectiveness of OLMA in addressing the above two challenges and the resulting improvement in forecasting accuracy. The results also indicate that the perspectives of entropy and frequency bias provide a new and feasible research direction for time series forecasting.

## 1 INTRODUCTION

Time series forecasting is an important fundamental technique with broad applications in energy management, financial trading, transportation optimization, weather prediction and healthcare monitoring. As the volume of temporal data continues to grow rapidly, enhancing forecasting accuracy has become an urgent need. As machine learning advances, neural networks have become the dominant approach for time series forecasting. Most research efforts have concentrated on developing increasingly sophisticated models to capture the underlying distributions of time series in real-world settings (Wang et al. (2025); Liu et al. (2024a; 2023); Wu et al. (2022)).

However, from a data-centric perspective, real-world time series are inevitably corrupted by purely random noise. This noise overlays the underlying learnable patterns, rendering perfect forecasting impossible, regardless of how strong the neural network's capacity to model the data distribution is. Fang et al. (2019); Rho (2020) have shown that the estimation error of a random variable (or stochastic process) has a theoretical lower bound, which is positively correlated with its own entropy. However, they have not further investigated whether the theoretical lower bounds of the estimation errors decrease when multiple correlated stochastic processes are present.

In this work, we provide a concrete result that there necessarily exists a unitary transformation that decreases the marginal entropy of multiple correlated Gaussian stochastic processes (the sum of the entropy of the individual processes). In Section 3, a detailed proof of this theorem is presented. By modeling the label data of time series as a combination of a learnable informative component and an unlearnable stochastic component, this conclusion provides theoretical guidance for reducing the lower bound of forecasting error. In particular, our experiments demonstrate that, in practical

scenarios, the DFT applied along the channel dimension serves as a unitary transformation that reduces entropy.

Another prevalent challenge in time series forecasting is the frequency bias of neural networks (Yu et al. (2024); Kiessling & Thor (2022); Cao et al. (2019)). More precisely, neural networks tend to exhibit inherent differences in their learning capacity in different frequency bands. In fact, this issue is not confined to the domain of time series forecasting, it also poses a significant challenge in the field of computer vision. Fang & Xu (2024) and Piao et al. (2024) have independently tackled the problem of frequency bias by introducing frequency domain transformation modules into their respective architectures.

To enhance the universality of using frequency domain transformations to alleviate the inherent frequency bias of neural networks, we embed the transformation directly into the loss function, enabling its application to any supervised learning method without altering the target network. Specifically, inspired by Neelamani et al. (2004), we applied the DFT and DWT to the temporal dimension of time series labels and predictions.

In summary, we propose a novel supervision method for time series forecasting, termed OLMA, which applies frequency domain transformations to both the channel and temporal dimensions of multivariate time series. This approach not only reduces the entropy of label noise, but also mitigates the inherent frequency bias of neural networks. Since this solution is formulated as a loss function, it can be seamlessly integrated into any supervised model. The contributions of this paper are summarized below.

- We analyze time series forecasting errors from the perspective of entropy, then we theoretically and empirically demonstrate that there exists a unitary transformation that reduces the marginal entropy of multivariate correlated Gaussian processes. Moreover, it has been validated that constructing loss in the frequency domain along the temporal dimension can alleviate the frequency bias of neural networks.

- We propose OLMA, a supervision method that applies frequency domain loss along both the channel and temporal dimensions of time series. OLMA provides a minimalist yet effective approach to reducing the entropy of label noise while mitigating the inherent frequency bias of neural networks. Moreover, it is plug-and-play and can be seamlessly integrated into any supervised learning framework.

- On 9 public time series forecasting datasets, OLMA was evaluated with multiple representative baseline models and demonstrated superior performance compared to their original time domain supervision methods. Our work calls for time series forecasting research not only to pursue innovations in model architectures but also to devote greater attention to the intrinsic properties of data, in order to discover more efficient and generalizable approaches for improving forecasting accuracy.

## 2 RELATED WORKS

**Time series forecasting approaches.** With the rise of neural networks, time series modeling had significantly evolved, particularly with the advent of recurrent neural network (RNN)-based methods (e.g., DeepAR Salinas et al. (2020), LSTNet Li et al. (2020), DA-RNN Qin et al. (2017)) and convolutional neural network (CNN)-based approaches (e.g., TCN Bai et al. (2018), SCINet Liu et al. (2022), TimesNet Wu et al. (2022)). The introduction of the Transformer Vaswani et al. (2017) architecture, known for its exceptional modeling capacity, had led to a surge in Transformer-based forecasting models. Early examples included InformerZhou et al. (2021), which applied Transformers directly to time series forecasting; PatchTST Nie et al. (2022), which treated time series segments as tokens; and iTransformer Liu et al. (2023), which integrated both temporal and channel-wise dependencies. Interestingly, DLinear Zeng et al. (2023) demonstrated the surprising effectiveness of simple linear layers in time series forecasting, prompting the development of multilayer perceptron (MLP)-based time domain models such as TimeXer Wang et al. (2024b), TimeMixer Wang et al. (2024a), and WPMixer Murad et al. (2025). Furthermore, TimeLLM Jin et al. (2023), AutoTime Liu et al. (2024b), and TimeCMA Liu et al. (2024a) proved the effectiveness of large language models (LLMs) in time series forecasting. Recently, the Mamba-based model, S-Mamba Wang et al.

(2025) and Affirm Wu et al. (2025), had also demonstrated the superior capabilities of state space models in time series forecasting.

**Forecasting errors from entropy perspective.** Fang et al. (2019) established information-theoretic bounds on estimation and forecasting errors in time series, showing their dependence on the conditional entropy of the data. Rho (2020) proposed a framework to evaluate time series forecasting algorithms by relating lower bounds of forecasting error to the conditional entropy rate of the series. Both suggested that the lower bound of time series forecasting error was positively correlated with the entropy of the labels, offering an insightful perspective. Nevertheless, they did not explore how decreasing information entropy could enhance forecasting performance.

**Frequency bias of neural networks.** Cao et al. (2019) and Kiessling & Thor (2022) had rigorously demonstrated that neural networks exhibit frequency bias. Fang & Xu (2024) tackled the frequency bias of deep neural networks by using a frequency-based multi-grade learning approach to better capture high-frequency features. Tancik et al. (2020) addressed the frequency bias of MLPs by using Fourier feature mappings, enabling faster learning of high-frequency functions in low-dimensional tasks. Piao et al. (2024) proposed Fredformer to mitigate frequency bias by learning features evenly across all frequency bands, improving forecasting of high- and low-frequency components. These methods addressed frequency bias by designing network architectures that incorporate frequency domain transformations, but their applicability is often limited to specific models.

## 3 METHODOLOGY

This chapter first theoretically demonstrates the possibility of reducing the marginal entropy of multivariate time series (Section 3.1), and then presents the detailed formulation of the OLMA loss (Section 3.2).

### 3.1 THEORETICAL DERIVATION

**Preliminaries.** Let $x$ be a continuous random variable with differential entropy $h(x)$, and let $\hat{x}$ be an unbiased estimate of $x$ formed without any side information. Under this constraint, unbiasedness requires $\hat{x} = \mathbb{E}[x]$, so the estimation error $e = x - \hat{x} = x - \mathbb{E}[x]$ is zero-mean. Since differential entropy is translation-invariant, $h(e) = h(x)$. According to the maximum entropy theorem for continuous random variables with given mean and variance (Jaynes (1957)), for any random variable, its entropy is upper-bounded by that of a Gaussian with the same variance,

$$h(e) = h(x) \leq \frac{1}{2} \log(2\pi e \mathrm{Var}(e)), \tag{1}$$

where Var denotes the variance. It can be rearranged to give the desired lower bound on the mean squared error,

$$\mathbb{E}[(x - \hat{x})^2] = \mathrm{Var}(e) \geq \frac{1}{2\pi e} 2^{2h(x)}. \tag{2}$$

The equality holds if and only if $x$ is Gaussian (Fang et al. (2019)).

Let $Y \in \mathbb{R}^{c \times l}$ denote the time series labels with $c$ dimensions (channels) and length $l$. Followed by Li et al. (2022); Zhou et al. (2022); Box & Jenkins (1968), the label is decomposed into two components as $Y = Z + N$, where $Z, N \in \mathbb{R}^{c \times l}$ denote components of learnable deterministic process (without randomness, the entropy is theoretically zero) and components of unlearnable stochastic noise respectively. We assume that $N$ is Gaussian (Aigrain & Foreman-Mackey (2023); Yuan & Qiao (2024)) and mutually independent across different time steps for analytical tractability. Thus, the lower bound of $N_i \in \mathbb{R}^l$, the variable $i^{th}$ of $N$, is

$$\sum_{t=1}^{l} \mathbb{E}[(N_i[t] - \hat{N}_i[t])^2] \geq \sum_{t=1}^{l} \frac{1}{2\pi e} 2^{2h(N_i[t])} = \frac{l}{2\pi e} 2^{2h(N_i)}. \tag{3}$$

This indicates that the lower bound of the forecasting error for each time series variable is positively correlated with its own entropy. If there exists an invertible transformation that can reduce entropy, the lower bound of the forecasting error can be decreased, thereby improving the forecasting accuracy. In this regard, we propose **Theorem 1**, which demonstrates that such transformation indeed exists.

**Theorem 1.** *If multiple Gaussian stochastic processes are internally independent and identically distributed (i.i.d.) but exhibit correlations across processes, then there necessarily exists a unitary transformation that reduces their marginal entropy, i.e., the sum of the entropy of each individual process.*

Before proving Theorem 1, we state 3 lemmas that will be used in the proof.

**Lemma 1.** *Let $A \in \mathbb{C}^{n \times n}$ be a positive definite Hermitian matrix with main diagonal elements $a_{11}, a_{22}, \ldots, a_{nn}$. Then the determinant of $A$ satisfies the inequality:*

$$\det(A) \leq \prod_{j=1}^{n} a_{jj}, \tag{4}$$

*with equality if and only if $A$ is a diagonal matrix.*

**Proof of Lemma 1.** Since $A$ is positive definite Hermitian, it admits a unique Cholesky decomposition $A = LL^*$, where $L$ is a lower triangular matrix with $l_{ii} > 0$ for $i = 1, 2, \ldots, n$, and $L^*$ denotes the conjugate transpose of $L$ (Pedersen et al. (2024)). The determinant of $A$ can be expressed as

$$\det(A) = \det(LL^*) = |\det(L)|^2 = (\prod_{i=1}^{n} l_{ii})^2. \tag{5}$$

The diagonal elements of $A$ are given by $a_{ii} = \Sigma_{k=i}^{n} |l_{ik}|^2 \geq |l_{ii}|^2$, for $i = 1, 2, \ldots, n$. Taking the product of these inequalities yields

$$\prod_{i=1}^{n} a_{ii} \geq \prod_{i=1}^{n} |l_{ii}|^2 = (\prod_{i=1}^{n} l_{ii})^2 = \det(A). \tag{6}$$

Equality holds if and only if $a_{ii} = l_{ii}^2$ for all $i$, which requires $l_{ik} = 0$ for all $k < i$. This implies $L$ is diagonal, and consequently $A = LL^*$ is also diagonal. Thus, Lemma 1 is proved.

**Lemma 2** (Unitary diagonalization of a Hermitian matrix). *Let $A \in \mathbb{C}^{n \times n}$ be a Hermitian matrix (i.e., $A = A^*$). Then there exists a unitary matrix $U \in \mathbb{C}^{n \times n}$ (i.e., $U^* = U^{-1}$) and a real diagonal matrix $\Lambda = \operatorname{diag}(\lambda_1, \lambda_2, \ldots, \lambda_n)$ such that*

$$A = U\Lambda U^*. \tag{7}$$

*The columns of $U$ form an orthonormal basis of $\mathbb{C}^n$ consisting of eigenvectors of $A$, and the diagonal entries of $\Lambda$ are the corresponding eigenvalues. Furthermore, if $A$ is positive definite, then all eigenvalues $\lambda_i$ are positive (Cederbaum et al. (1989)).*

**Lemma 3** (Path-Connectedness of the Unitary Group). *The unitary group $\mathcal{U}(n)$ is path-connected. That is, for any two unitary matrices $U, V \in \mathcal{U}(n)$, there exists a continuous function $\varphi : [0, 1] \to \mathcal{U}(n)$ such that $\varphi(0) = U$ and $\varphi(1) = V$ (Knapp & Knapp (1996)).*

**Proof of Theorem 1.** Let $G \in \mathbb{R}^{c \times l}$ denote $c$ correlated Gaussian stochastic processes and length $l$. For each process $G_i$, since the variables are i.i.d. Gaussian, its entropy is

$$h(G_i) = \frac{1}{2} log((2\pi e)^l \det(\Sigma_i)) \overset{\text{i.i.d.}}{=} \frac{l}{2} log(2\pi e \sigma_i^2), \tag{8}$$

where $\Sigma_i$ is the covariance matrix of $G_i$, and $\sigma_i^2$ corresponds to the variance of each Gaussian random variable. The sum of the marginal entropy of $G$ is

$$\sum_{i=1}^{c} h(G_i) = \sum_{i=1}^{c} \frac{l}{2} log(2\pi e \sigma_i^2) = \frac{l}{2} log((2\pi e)^c \prod_{i}^{c} \sigma_i^2). \tag{9}$$

It is evident that $\prod_{i}^{c} \sigma_i^2$ is the product of the first-order principal minors of the covariance matrix of $G$, which is denoted as $S = \frac{1}{l} GG^*$, where $G^*$ is the conjugate transpose matrix of $G$. When a unitary transformation $F$ is applied to $G$, the covariance matrix $S_u$ is transformed into

$$S_u = \frac{1}{l} FG(FG)^* = F(\frac{1}{l} GG^*)F^* = FSF^*. \tag{10}$$

According to **Lemma 1**, $\det(S) < \prod_i^c \sigma_i^2$, because the $G_i$s are correlated, the equality does not apply. According to **Lemma 2**, there is necessarily a unitary transformation $F_v$ such that $S_u$ becomes a diagonal matrix, which means $\det(S_u) = \prod_{i=1}^c \hat{\sigma}_i^2$, where $\hat{\sigma}_i^2$ is the element on the main diagonal of $S_u$. Since a unitary transformation does not change the determinant of a matrix, $\det(S) = \det(S_u)$.

In summary, there exists a unitary transformation $\varphi(0) = I$ (identity matrix) such that the main diagonal of the covariance matrix of $G$ remains unchanged, and there also exists a unitary transformation $\varphi(1) = F_v$ that reduces it to its minimum value $\det(S)$. Since the unitary space is continuous (from $\varphi(0) = I$ to $\varphi(1) = F_v$), the range of attainable values forms a closed real value interval (from $\prod_i^c \sigma_i^2$ to $\det(S)$), there necessarily exists a $F_\lambda = \varphi(\lambda), 0 < \lambda < 1$ such that $\det(S) < \prod_{i=1}^c \hat{\sigma}_i^2 < \prod_i^c \sigma_i^2$, that is the product of the main diagonal entries is reduced. In conjunction with Eq. 9, it can be rigorously deduced that there necessarily exists a unitary transformation that reduces the marginal entropy of the Gaussian process. Thus, **Theorem 1** is proved.

## 3.2 OLMA Loss

The forecasting of the model is denoted as $\hat{Y} \in \mathbb{R}^{l \times c}$, and the corresponding label as $Y \in \mathbb{R}^{l \times c}$. According to **Theorem 1**, the DFT applied along the channel dimension acts as a unitary transformation that can reduce the marginal entropy of multivariate time series labels (the experimental validation is presented in Section 4). The computation can be explicitly formulated as

$$\mathcal{L}_{\text{olma}}^{(c)} = \alpha \sum_{t=0}^{l-1} \left\| F_f(\hat{Y}_{t,:}) - F_f(Y_{t,:}) \right\|_1,  \tag{11}$$

where $0 < \alpha < 1$ is the hyperparameter to adjust the strength of $\mathcal{L}_{\text{olma}}^{(c)}$, $\hat{Y}_{t,:}$ and $Y_{t,:}$ are forecasting and label sequence of the $t^{th}$ time step respectively and $F_f$ represents DFT that detailed calculation is

$$F_f(Y_{t,:})[k] = \sum_{n=0}^{c-1} Y_{t,n} \cdot e^{-2\pi \mathrm{i} k n / c}, \quad k = 0, 1, \dots, c-1,  \tag{12}$$

where $\mathrm{i}$ is the imaginary unit.

To alleviate the frequency bias of neural networks, we also apply frequency domain transformations directly at the supervision stage. This provides the most convenient way to adapt to all supervised time series forecasting models. Inspired by Neelamani et al. (2004), we perform DFT and DWT along the temporal dimension of the time series. Applying a full DFT to long non-stationary signals may yield misleading frequency representations, since it assumes global stationarity and overlooks localized variations. In contrast, Wavelet Transform, a localized alternative to the short-time Fourier Transform, captures both temporal and frequency information, making it effective for modeling long-term non-stationary patterns in time series. The computation of $\mathcal{L}_{\text{olma}}^{(t)}$ is

$$\mathcal{L}_{\text{olma}}^{(t)} = \beta \sum_{i=0}^{c-1} \left\| F_f(\hat{Y}_{:,i}) - F_f(Y_{:,i}) \right\|_1 + \gamma \sum_{i=0}^{c-1} \left\| F_w(\hat{Y}_{:,i}) - F_w(Y_{:,i}) \right\|_1,  \tag{13}$$

where $\hat{Y}_{:,i}$ and $Y_{:,i}$ are forecasting and label sequence of the $i^{th}$ channel respectively, the hyperparameters $\beta$ and $\gamma$ (where $0 < \beta, \gamma < 1$ and $\alpha + \beta + \gamma = 1$) are introduced to adjust the strength of alignment in the Fourier and Wavelet domains, respectively, and $F_w$ denotes the DWT. For $k = 1, 2, \dots, l/2$, there are

$$Y_{2k-1,i} = \frac{cA_k + cD_k}{\sqrt{2}}, \quad Y_{2k,i} = \frac{cA_k - cD_k}{\sqrt{2}}, \quad F_w(Y_{:,i}) = \{cA_1, \dots, cA_k, cD_1, \dots, cD_k\}.  \tag{14}$$

where $cA$ is the approximation coefficient and $cD$ is the detail coefficient of $Y_{:,i}$. Note that squared or higher-order norms for the error are not adopted. Because, in most time series data, the magnitude of frequency components varies significantly across different bands in the frequency domain. In particular, low-frequency components typically dominate and exhibit much larger amplitudes than high-frequency components. To ensure stability of the loss, the L1 norm is adopted. Finally, the OLMA loss $\mathcal{L}_{\mathcal{O}}$ is defined as a linear combination of the frequency domain losses along the temporal and channel dimensions,

$$\mathcal{L}_{\mathcal{O}} = \mathcal{L}_{\text{olma}}^{(t)} + \mathcal{L}_{\text{olma}}^{(c)}.  \tag{15}$$

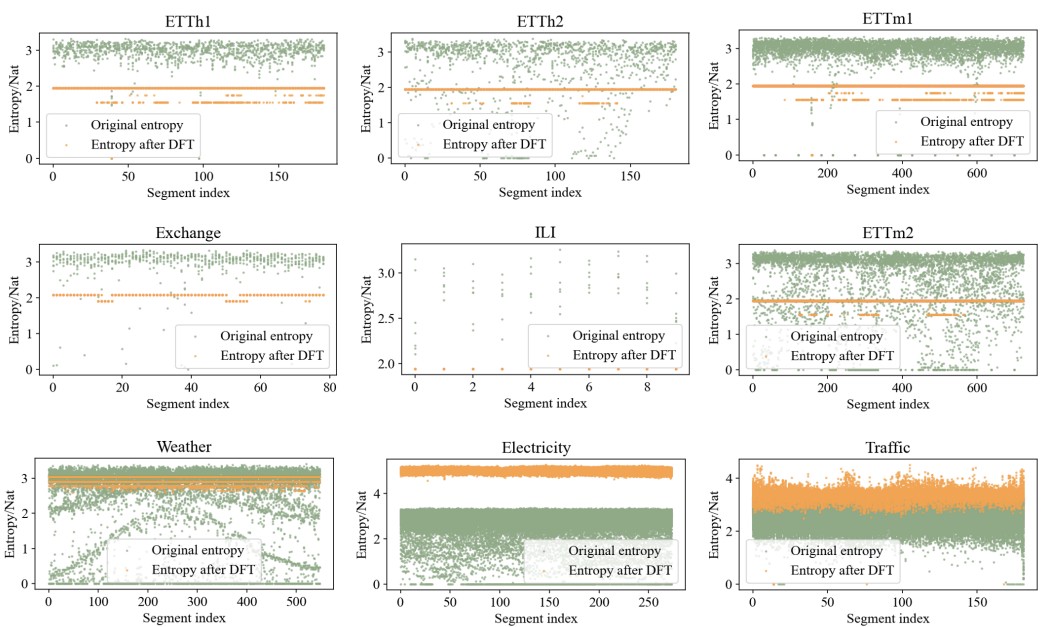

Figure 1: Entropy changes after applying channel-wise DFT in different time series datasets.

## 4 EXPERIMENTS

### 4.1 LOW ENTROPY REPRESENTATION OF TIME SERIES

Inspired by **Theorem 1**, we aim to develop a representation method that reduces the marginal entropy of time series along the temporal dimension. Since the DFT decomposes a sequence into different frequency components, we apply it along the channel dimension so that energy from the same frequency band is concentrated within the same channel. This reduces the uncertainty within each individual channel and thereby decreases the entropy of the time series. For the original real-valued time series $Y_{:,i} \in \mathbb{R}^l$, we compute its Shannon entropy. Because the true probability distribution of each value is inaccessible, we replace it with the empirical probability estimated from the data. Concretely, the values of $Y_{:,i}$ are first partitioned into $M$ equal-width, non-overlapping bins. Let $n_k = \textbf{number}(Y_{j,i} \in M_k), j = 0, 1, \ldots l-1$ denote the number of series points that fall into $M_k$, the $k^{th}$ interval of $M$. The empirical probability $p_k = n_k/l$. Therefore, the Shannon entropy of $Y_{:,i}$ can be expressed as

$$H(Y_{:,i}) = -\sum_{k=1}^{M} p_k log(p_k), \tag{16}$$

Since $Y_{:,i}$ becomes a complex-valued sequence after the DFT, we treat it as a two-dimensional discrete sequence and compute its joint entropy following the method described above. Followed by Wang et al. (2025); Liu et al. (2023); Wu et al. (2022), ETT (4 subsets), Exchange, Illness (ILI), Weather, Electricity (ECL), Traffic datasets are used in our experiments (see Appendix A.1 for dataset details). Each dataset is segmented into 96-length segments along the temporal dimension. As shown in Figure1, the entropy of each segment is indicated with a scatter plot, where green represents the entropy of the original sequence and orange represents the entropy after applying DFT along the channel dimension. Evidently, in most scenarios, representing time series using DFT along the channel dimension can significantly reduce their marginal entropy, which experimentally validates **Theorem 1**. Moreover, this representation significantly reduces the entropy differences across different time series samples, which indicates a more uniform distribution of information, without extreme redundancy or uncertainty. However, for a few datasets, such as ECL, this can lead to an increase in entropy, which may affect the forecasting performance of certain models (a detailed discussion is provided in Section 4.3 and 4.4).

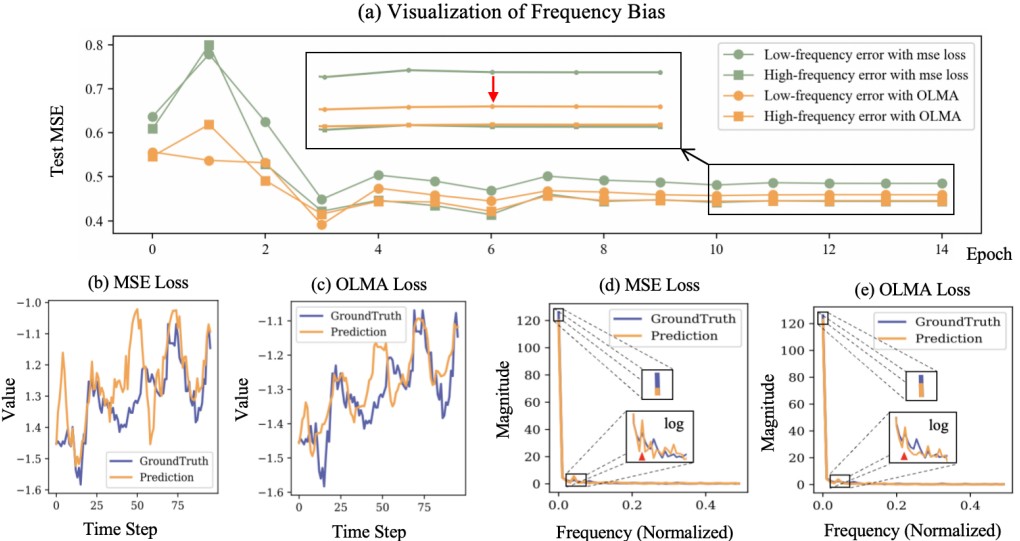

Figure 2: (a) denotes the forecasting error across different frequency bands on the ETTh1 test set during training, reflecting the frequency bias of the network. (b) and (d) visualize the forecasting and ground truth values in the time and frequency domains under time domain MSE supervision, respectively. (c) and (e) visualize the forecasting and ground truth values in the time and frequency domains under OLMA supervision, respectively.

## 4.2 ALLEVIATION OF FREQUENCY BIAS

Followed by Yu et al. (2024), we quantify frequency bias by measuring the forecasting errors of different frequency bands. As evidenced by the two green curves in Figure 2 (a), the model manifests a pronounced frequency bias, exhibiting a preferential tendency toward capturing high-frequency components. It is worth noting that the DLinear model employed in this experiment was specifically designed to balance low- and high-frequency learning through parallel seasonal and trend branches, yet the issue of frequency bias still persists. After applying OLMA supervision, the model's ability to learn low-frequency components is substantially enhanced, while its ability to capture high-frequency components remains largely unaffected. This provides empirical evidence that applying supervision in the frequency domain allows the network to access information across all frequency bands more directly, effectively alleviating its intrinsic frequency bias.

For greater clarity, we visualize the forecasting in both the time and frequency domains. As illustrated in Figure 2 (b), the ground truth exhibits an overall upward trend, which is manifested primarily in the low-frequency components. However, under time domain supervision, the network exhibits limited capacity in capturing low-frequency information, and consequently, such a trend cannot be adequately fitted. In contrast, under the guidance of OLMA, the network exhibits a markedly improved capacity to approximate the trend component, as shown in the plot (c). In addition, comparison of plots (d) and (e) reveals that OLMA supervision provides a more faithful approximation of the primary and secondary spectral peaks in the low-frequency band than conventional time domain MSE. This serves as compelling evidence for the network's enhanced proficiency in modeling low-frequency structures, substantiating the claim that direct frequency domain supervision provides a principled solution to alleviate frequency bias. In addition, we also discuss the issue of frequency bias from the perspective of the data, see Appendix A.2 for details.

## 4.3 PERFORMANCE OF OLMA

We further validate the effectiveness of OLMA by incorporating it into several state-of-the-art baseline models across diverse settings. These methods include Mamba-based S-Mamba Wang et al. (2025), LLM-based TimeCMA Liu et al. (2024a), Transformer-based iTransformer Liu et al.

Table 1: Performance of OLMA on different time series datasets. Lower forecasting errors indicate better performance. The best results are highlighted in bold. TDL denotes the temporal domain loss (MSE) corresponding to each baseline.

| Dataset | Loss | S-Mamba MSE | MAE | TimeCMA MSE | MAE | iTransformer MSE | MAE | TimesNet MSE | MAE | TimeXer MSE | MAE | TimeMixer MSE | MAE | DLinear MSE | MAE |
|---|---|---|---|---|---|---|---|---|---|---|---|---|---|---|---|
| ETTh1 | TDL | 0.455 | 0.450 | 0.438 | 0.441 | 0.454 | 0.448 | 0.460 | 0.455 | 0.437 | 0.437 | 0.447 | 0.440 | 0.423 | 0.437 |
| | OLMA | **0.432** | **0.426** | **0.433** | **0.434** | **0.444** | **0.437** | **0.445** | **0.443** | **0.436** | **0.429** | **0.435** | **0.429** | **0.413** | **0.424** |
| ETTh2 | TDL | 0.381 | 0.405 | 0.407 | 0.420 | 0.383 | 0.407 | 0.407 | 0.421 | 0.368 | 0.396 | 0.374 | 0.401 | 0.431 | 0.447 |
| | OLMA | **0.362** | **0.391** | **0.388** | **0.408** | **0.376** | **0.400** | **0.401** | **0.416** | **0.363** | **0.389** | **0.368** | **0.394** | **0.415** | **0.434** |
| ETTm1 | TDL | 0.398 | 0.405 | 0.393 | 0.406 | 0.407 | 0.410 | 0.411 | 0.418 | 0.382 | 0.397 | 0.381 | 0.396 | 0.357 | 0.379 |
| | OLMA | **0.379** | **0.386** | **0.383** | **0.391** | **0.397** | **0.398** | **0.393** | **0.402** | **0.377** | **0.385** | **0.378** | **0.385** | **0.353** | **0.372** |
| ETTm2 | TDL | 0.288 | 0.332 | 0.290 | 0.333 | 0.283 | 0.332 | 0.296 | 0.332 | 0.274 | 0.322 | 0.275 | 0.323 | 0.267 | 0.332 |
| | OLMA | **0.278** | **0.319** | **0.285** | **0.323** | **0.283** | 0.324 | **0.285** | **0.323** | **0.271** | **0.315** | **0.273** | **0.319** | **0.263** | **0.322** |
| Weather | TDL | 0.251 | 0.276 | 0.248 | 0.281 | 0.258 | 0.278 | 0.259 | 0.286 | 0.241 | 0.271 | 0.240 | 0.272 | 0.246 | 0.300 |
| | OLMA | **0.241** | **0.265** | **0.245** | **0.275** | **0.255** | **0.275** | **0.257** | **0.281** | **0.239** | **0.266** | **0.242** | **0.266** | **0.240** | **0.280** |
| Exchange | TDL | 0.367 | 0.408 | 0.446 | 0.457 | 0.360 | 0.403 | 0.408 | 0.439 | 0.372 | 0.409 | 0.352 | 0.398 | 0.367 | 0.416 |
| | OLMA | **0.350** | **0.398** | **0.416** | **0.441** | **0.353** | **0.401** | **0.403** | **0.434** | **0.349** | **0.398** | **0.342** | **0.393** | **0.315** | **0.394** |
| ILI | TDL | 2.027 | 1.066 | 1.864 | 0.873 | 2.552 | 1.109 | 2.263 | 0.928 | 2.143 | 0.961 | 2.088 | 0.977 | 2.169 | 1.041 |
| | OLMA | **1.806** | **0.853** | **1.858** | **0.869** | **2.516** | **1.097** | **2.045** | **0.869** | **2.124** | **0.944** | **1.739** | **0.828** | **2.049** | **0.970** |
| ECL | TDL | 0.170 | 0.265 | 0.213 | 0.307 | 0.178 | 0.270 | 0.194 | 0.296 | **0.171** | 0.270 | **0.182** | 0.273 | **0.166** | 0.264 |
| | OLMA | **0.167** | **0.262** | **0.200** | **0.296** | **0.169** | **0.258** | **0.188** | **0.288** | 0.172 | **0.268** | 0.183 | **0.272** | 0.167 | **0.263** |
| Traffic | TDL | 0.414 | 0.276 | 0.697 | **0.370** | 0.428 | 0.282 | 0.625 | 0.331 | **0.466** | 0.287 | 0.499 | 0.322 | 0.434 | 0.295 |
| | OLMA | **0.412** | **0.265** | **0.696** | **0.370** | **0.421** | **0.270** | **0.616** | **0.319** | 0.468 | **0.277** | **0.496** | **0.308** | **0.433** | **0.293** |

(2023), CNN-based TimesNet Wu et al. (2022), linear-based DLinear Zeng et al. (2023), and MLP-based TimeMixer Wang et al. (2024a) and TimeXer Wang et al. (2024b). The average forecast errors of four horizons $\{96, 192, 336, 720\}$ for different methods on different datasets (ILI are $\{12, 24, 48, 96\}$) are shown in Table 1 (complete experimental results and detailed setting of hyperparameters are provided in the Appendix A.3). In accordance with commonly adopted protocols, each dataset is divided into training (60%), validation (20%) and test (20%) subsets. The experimental results indicate that OLMA, when directly integrated into diverse baseline models, consistently outperforms the widely adopted time domain supervision approaches. More intriguingly, OLMA eliminates the reliance on time domain supervision altogether, instead representing time series labels purely within the frequency domain. This phenomenon can be explained by Parseval's Theorem Folland (2009).

**Parseval's Theorem.** Let $x(t)$ be the time domain signal of interest. The total energy of the signal in the time domain is equal to that in the frequency domain. Mathematically, this relationship is expressed as

$$\int_{-\infty}^{\infty} |x(t)|^2 \, dt = \int_{-\infty}^{\infty} |X(f)|^2 \, df \tag{17}$$

where $X(f)$ is the Fourier Transform of $x(t)$. This indicates that the frequency domain representation of a signal preserves its total energy and only redistributes it across frequency components. Therefore, applying supervision in the frequency domain does not result in any energy loss, and retains the full informational content of the original signal. Thus, combining time domain supervision with OLMA does not provide any additional information gain (experimental validation is provided in the Appendix A.4).

Consequently, OLMA constitutes an information-lossless representation of time series that effectively reduces their intrinsic disorder, as measured by entropy. Nevertheless, as shown in Figure 1, for datasets characterized by more intricate channel interactions, exemplified by ECL, the Fourier Transform can inadvertently increase the entropy of the time series. The performance of methods such as DLinear and TimeMixer on the ECL, as reported in Table 1, substantiates this finding. Because their architectures are relatively simple, these models are unable to counteract the increase in disorder induced by entropy growth, resulting in limited performance improvement.

## 4.4 ABLATIONS

A detailed ablation study is conducted on the ETTh1 and ECL dataset using iTransformer and TimeMixer to examine the contributions of two frequency domain loss components in OLMA, those

Table 2: Ablation study of OLMA on channel and temporal losses.

| Channel | Temporal | iTransformer | | | | TimeMixer | | | |
|---|---|---|---|---|---|---|---|---|---|
| | | ETTh1 | | ECL | | ETTh1 | | ECL | |
| ✗ | ✗ | 0.454 | 0.448 | 0.178 | 0.270 | 0.447 | 0.440 | 0.182 | 0.273 |
| ✓ | ✗ | 0.448 | 0.440 | 0.172 | 0.263 | 0.439 | 0.433 | 0.197 | 0.284 |
| ✗ | ✓ | 0.451 | 0.444 | 0.175 | 0.262 | 0.442 | 0.436 | 0.183 | 0.274 |
| ✓ | ✓ | 0.444 | 0.437 | 0.169 | 0.258 | 0.435 | 0.429 | 0.183 | 0.272 |

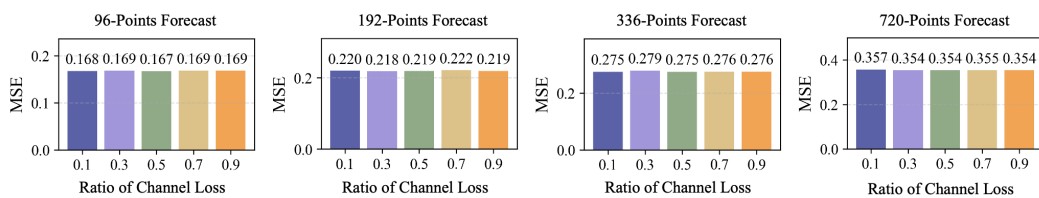

Figure 3: Impact of the ratio between channel and temporal losses in OLMA on forecasting error.

are the channel-wise $\mathcal{L}_{\mathrm{olma}}^{(c)}$ and the temporal-wise $\mathcal{L}_{\mathrm{olma}}^{(t)}$. In Table 2, "Channel" represents $\mathcal{L}_{\mathrm{olma}}^{(c)}$ and "Temporal" represents $\mathcal{L}_{\mathrm{olma}}^{(t)}$. Both are discarded, denoting the MSE loss originally used by the models. For dataset such as ETTh1, where channel-wise DFT effectively reduces information entropy, iTransformer and TimeMixer achieve enhanced forecasting performance by leveraging solely $\mathcal{L}_{\mathrm{olma}}^{(c)}$. However, for dataset like ECL, where channel-wise DFT increases entropy, MLP-based predictors such as TimeMixer are substantially affected, whereas Transformer-based models like iTransformer remain largely unaffected. Moreover, the stabilization of entropy distribution further enhances iTransformer's forecasting performance. This further corroborates the analysis presented in Section 4.1.

### 4.5 IMPACT OF CHANNEL AND TEMPORAL LOSSES ON FORECASTING PERFORMANCE

In the ablation study, we have already demonstrated that jointly applying losses along both the channel and temporal dimensions yields superior forecasting performance. However, an open question remains that does the relative weighting between the two losses exert a significant influence on forecasting accuracy? To this end, we take the Weather dataset as an example and conduct detailed experiments using iTransformer. Specifically, as shown in Figure 3, we vary the proportion of the channel loss across {0.1, 0.3, 0.5, 0.7, 0.9}, and evaluate the model under four different forecasting lengths {96, 192, 336, 720}. It is evident that even under substantial variations in the relative weighting of channel and temporal losses, the model's forecasting performance remains largely unaffected, which implies that within a relatively wide range of weight assignments, the model forecasting performance remains stable and strong, eliminating the need for tedious and expensive hyperparameter fine-tuning. Additional experiments are provided in the Appendix A.5.

## 5 CONCLUSIONS AND FUTURE DIRECTIONS

**Conclusions.** We prove that unitary transformations can reduce the marginal entropy of multivariate time series, yielding low-entropy representations that enhance forecasting accuracy. Meanwhile, we mitigate frequency bias of neural networks by enforcing supervision directly in the frequency domain. As a combination of these two solutions, OLMA provides a minimalist approach that can be seamlessly integrated into any supervised learning model.

**Future directions.** We reveal two overlooked issues that offer valuable guidance for future research. Firstly, we analyze time series representations from the perspective of entropy. Although we have proposed an effective representation for entropy reduction in time series, this approach still leaves considerable room for improvement. Future work should strive to identify representations with minimal entropy in order to further lower the fundamental bound of forecasting error. Secondly, future work should assess model performance across different frequency bands in time series and develop more targeted solutions accordingly.

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

Table 3: Information of each dataset. Channel represents the variate number of each dataset. Length is the total number of time steps. Sampling rate denotes the sampling interval of time steps. Domain refers to the application area to which the dataset belongs.

| Dataset | Channel | Length | Sampling rate | Domain |
|---------|---------|--------|---------------|--------|
| ETTh1&ETTh2 | 7 | 17420 | 1 Hour | Energy |
| ETTm1&ETTm2 | 7 | 69680 | 15 Minutes | Energy |
| Weather | 21 | 52696 | 10 Minutes | Climate |
| Exchange | 8 | 7588 | 1 Day | Finance |
| ILI | 7 | 966 | 1 Week | Healthcare |
| Electricity | 321 | 26304 | 1 Hour | Energy |
| Traffic | 862 | 17544 | 1 Hour | Transportation |

# A  APPENDIX

## A.1  DATASETS

The datasets used in our experiments span a wide variety of real-world time series applications. The ETT dataset collects industrial temperature and torque data, divided into four subsets (ETTh1, ETTh2, ETTm1, ETTm2), each reflecting different temporal granularities and periods for evaluating long-sequence forecasting models. The Weather dataset consists of meteorological variables such as temperature, humidity, and wind speed across multiple geographic locations, and is widely used in environmental forecasting. The Exchange Rate dataset contains foreign exchange rates of eight major currencies against the US dollar and is commonly used for financial time series forecasting. The ILI dataset comprises historical weekly records of flu-related case counts released by the United States Centers for Disease Control and Prevention, suitable for epidemiological modeling. The Electricity dataset reflects household-level electricity consumption across hundreds of clients and supports studies on energy demand forecasting. The Traffic dataset captures vehicle road occupancy across California's highway system, useful for urban mobility prediction. More detailed information about the sequence length, number of channels, and sampling rate for each dataset is provided in Table 3.

## A.2  EXPLAINING FREQUENCY BIAS FROM DATA PERSPECTIVE

The inherent frequency preference of neural networks is a well-recognized phenomenon. However, the characteristics of the data itself can also influence the network's ability to learn across different frequency components. In the time domain, strong correlations between time points (e.g., high values of the autocorrelation function) imply that the errors of adjacent points in the loss computation are highly correlated. This, in turn, biases gradient descent updates toward capturing local variation patterns. As illustrated in Figure 2 (b), the time series exhibits strong local oscillations, indicating that high-frequency components dominate within these regions. This provides an additional explanation for why the model tends to prioritize learning high-frequency information. In the frequency domain, after applying the Fourier transform, different frequency components become approximately orthogonal (i.e., weakly correlated). This implies that the error associated with each frequency component contributes independently to the loss function. Consequently, gradient descent updates the network parameters corresponding to each frequency in an independent manner, preventing any single frequency from dominating the optimization process. To validate this, we conducted experiments on the ETTh1 dataset. As shown in Figure 4 (a), the data exhibits strong correlations among adjacent points in the time domain. In contrast, (b) and (c) clearly demonstrate that such correlations become much weaker in the frequency domain.

Specifically, we employ a Double Machine Learning (DML)Chernozhukov et al. (2018) approach to eliminate the influence of confounders on correlation estimation. The specific computation procedure is as Algorithm 1, which estimates the causal correlation from time point $t$ to $t'$ within a time series using residual regression based on the DML framework.

Let the input time series be $\{x_1, x_2, \ldots, x_N\}$, where $N$ is the total sequence length. It takes four parameters: a window size $w$ (is set to 2 in this work) to define the set of confounders (assuming that confounding factors exist in the sequence near the source time steps), the source and target time steps $t$ and $t'$ such that $t < t'$, and the maximum length used for computing sequence correlation $T_{\text{vis}}$

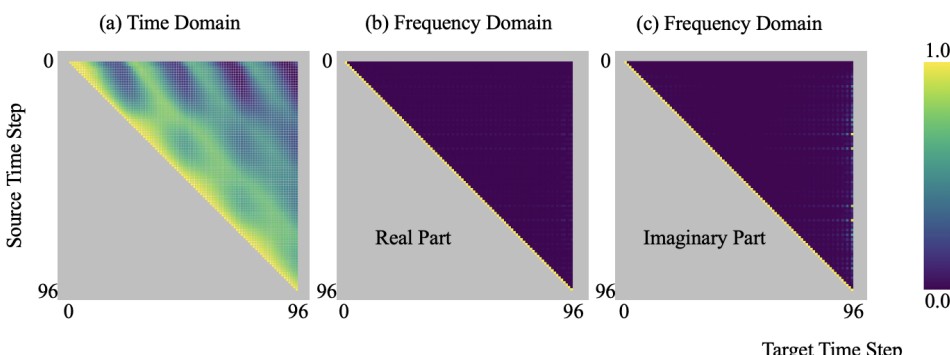

Figure 4: The correlation matrices on the ETTh1 dataset. (a) Correlation matrix in the time domain. (b) Correlation matrix of the real part after Fourier Transform. (c) Correlation matrix of the imaginary part.

(is set to 96 in this work). Each time point $i$ (from $w$ to $N - T_{\text{vis}}$) defines a local context for causal evaluation. The value at index $j = i + t$ serves as the source time steps set $T$, and the value at index $k = i + t'$ serves as the target time steps set $O$. For each such pair, we construct a set of confounders $C$ by collecting the $2w$ neighbors around $x_i$, excluding $x_i$ itself (and optionally excluding $x_j$ if it appears in the context window).

After collecting the samples of treatment, outcome, and confounders, we train two separate regression models:

- $\text{Model}_t$ to predict the source $T$ from the confounders $C$, and compute the residuals $\tilde{t}$.

- $\text{Model}_o$ to predict the target $O$ from the confounders $C$, and compute the residuals $\tilde{o}$.

These residuals represent the parts of the source and target that are not explained by the confounders. The final causal effect estimate $e_{t \rightarrow t'}$ is computed as the absolute value of the regression coefficient between $\tilde{t}$ and $\tilde{o}$, which is equivalent to the normalized covariance: $e_{t \rightarrow t'} = \left| \frac{\text{Cov}(\tilde{t}, \tilde{o})}{\text{Var}(\tilde{t})} \right|$, where Cov denotes the covariance calculation, and Var denotes the variance calculation. This value captures the residual dependence between the source and the target after adjusting for confounding variables, and thus serves as a proxy for causal correlation. Likewise, applying this algorithm in the frequency domain only requires a preliminary fixed-length Fourier Transform on the entire time series.

---

**Algorithm 1** Causal Correlation of Time Series

**Require:** Time series $\{x_1, x_2, \ldots, x_N\}$, window size $w$, offsets $t < t'$, visible range $T_{\text{vis}}$
**Ensure:** Estimated causal effect $e_{t \rightarrow t'}$
  1: Precompute confounders $C_i \leftarrow \{x_{i-w}, \ldots, x_{i-1}, x_{i+1}, \ldots, x_{i+w}\}$
  2: Initialize sample list $T \leftarrow [], O \leftarrow [], C \leftarrow []$
  3: **for** each $i = w$ to $N - T_{\text{vis}}$ **do**
  4:    $j \leftarrow i + t, \quad k \leftarrow i + t'$
  5:    **if** $k \geq N$ **then**
  6:      continue
  7:    **end if**
  8:    $T \leftarrow x_j, O \leftarrow x_k, C \leftarrow C_i$
  9: **end for**
10: Training a double machine learning model.
11: Train $\text{Model}_t(C, T)$, get residuals $\tilde{t} = \text{Model}_t(C) - T$
12: Train $\text{Model}_o(C, O)$, get residuals $\tilde{o} = \text{Model}_o(C) - O$
13: Compute $e_{t \rightarrow t'} = \left| \frac{\text{Cov}(\tilde{t}, \tilde{o})}{\text{Var}(\tilde{t})} \right|$
14: **return** $e_{t \rightarrow t'}$

---

Table 4: Full results of forecasting errors of OLMA on mainstream model baselines (TimeCMA, S-Mamba, iTransformer and TimesNet) which supervised by time domain losses (TDL) on different datasets. Lower values indicate better performance. The best results are highlighted in bold.

| Dataset | Length | TimeCMA TDL MSE | TDL MAE | OLMA MSE | OLMA MAE | S-Mamba TDL MSE | TDL MAE | OLMA MSE | OLMA MAE | iTransformer TDL MSE | TDL MAE | OLMA MSE | OLMA MAE | TimesNet TDL MSE | TDL MAE | OLMA MSE | OLMA MAE |
|---|---|---|---|---|---|---|---|---|---|---|---|---|---|---|---|---|---|
| ETTh1 | 96 | 0.392 | 0.413 | **0.389** | **0.406** | 0.386 | 0.405 | **0.368** | **0.387** | 0.386 | 0.405 | **0.382** | **0.396** | 0.389 | 0.412 | **0.382** | **0.403** |
| | 192 | 0.432 | 0.435 | **0.431** | **0.428** | 0.443 | 0.437 | **0.422** | **0.417** | 0.441 | 0.436 | **0.435** | **0.427** | 0.439 | 0.442 | **0.432** | **0.435** |
| | 336 | 0.467 | 0.452 | **0.464** | **0.445** | 0.489 | 0.468 | **0.466** | **0.438** | 0.487 | 0.458 | **0.483** | **0.454** | 0.494 | 0.471 | **0.480** | **0.458** |
| | 720 | 0.461 | 0.465 | **0.449** | **0.456** | 0.502 | 0.489 | **0.470** | **0.463** | 0.503 | 0.491 | **0.476** | **0.472** | 0.516 | 0.494 | **0.485** | **0.476** |
| | Avg | 0.438 | 0.441 | **0.433** | **0.434** | 0.455 | 0.450 | **0.432** | **0.426** | 0.454 | 0.448 | **0.444** | **0.437** | 0.460 | 0.455 | **0.445** | **0.443** |
| ETTh2 | 96 | 0.326 | 0.364 | **0.315** | **0.357** | 0.296 | 0.348 | **0.281** | **0.333** | 0.297 | 0.349 | **0.294** | **0.343** | 0.337 | 0.370 | **0.312** | **0.350** |
| | 192 | 0.419 | 0.420 | **0.382** | **0.398** | 0.376 | 0.396 | **0.357** | **0.382** | 0.380 | 0.400 | **0.373** | **0.391** | 0.405 | 0.415 | **0.392** | **0.406** |
| | 336 | 0.443 | 0.443 | **0.432** | **0.435** | 0.424 | 0.431 | **0.403** | **0.418** | 0.428 | 0.432 | **0.417** | **0.426** | 0.453 | 0.450 | **0.438** | **0.445** |
| | 720 | 0.441 | 0.453 | **0.423** | **0.442** | 0.426 | 0.444 | **0.408** | **0.432** | 0.427 | 0.445 | **0.420** | **0.439** | **0.434** | **0.448** | 0.460 | 0.464 |
| | Avg | 0.407 | 0.420 | **0.388** | **0.408** | 0.381 | 0.405 | **0.362** | **0.391** | 0.383 | 0.407 | **0.376** | **0.400** | 0.407 | 0.421 | **0.401** | **0.416** |
| ETTm1 | 96 | 0.324 | 0.365 | **0.313** | **0.352** | 0.333 | 0.368 | **0.311** | **0.348** | 0.334 | 0.368 | **0.325** | **0.359** | 0.334 | 0.375 | **0.326** | **0.360** |
| | 192 | 0.374 | 0.394 | **0.362** | **0.377** | 0.376 | 0.390 | **0.357** | **0.372** | 0.377 | 0.391 | **0.373** | **0.381** | 0.408 | 0.414 | **0.390** | **0.397** |
| | 336 | 0.407 | 0.415 | **0.396** | **0.399** | 0.408 | 0.413 | **0.393** | **0.395** | 0.426 | 0.420 | **0.411** | **0.407** | 0.415 | 0.422 | **0.402** | **0.410** |
| | 720 | 0.469 | 0.448 | **0.462** | **0.437** | 0.475 | 0.448 | **0.455** | **0.430** | 0.491 | 0.459 | **0.479** | **0.445** | 0.485 | 0.461 | **0.454** | **0.439** |
| | Avg | 0.393 | 0.406 | **0.383** | **0.391** | 0.398 | 0.405 | **0.379** | **0.386** | 0.407 | 0.410 | **0.397** | **0.398** | 0.411 | 0.418 | **0.393** | **0.402** |
| ETTm2 | 96 | 0.182 | 0.263 | **0.175** | **0.255** | 0.179 | 0.263 | **0.171** | **0.250** | 0.180 | 0.264 | **0.177** | **0.256** | 0.189 | 0.266 | **0.178** | **0.255** |
| | 192 | 0.257 | 0.316 | **0.245** | **0.298** | 0.250 | 0.309 | **0.238** | **0.295** | 0.250 | 0.309 | **0.243** | **0.300** | 0.252 | 0.307 | **0.246** | **0.301** |
| | 336 | 0.310 | 0.348 | **0.307** | **0.338** | 0.312 | 0.349 | **0.300** | **0.336** | 0.311 | 0.348 | **0.306** | **0.340** | 0.323 | 0.350 | **0.306** | **0.337** |
| | 720 | **0.412** | 0.404 | 0.414 | **0.400** | 0.411 | 0.406 | **0.402** | **0.396** | 0.412 | 0.407 | **0.407** | **0.399** | 0.419 | 0.405 | **0.411** | **0.398** |
| | Avg | 0.290 | 0.333 | **0.285** | **0.323** | 0.288 | 0.332 | **0.278** | **0.319** | 0.288 | 0.332 | **0.283** | **0.324** | 0.296 | 0.332 | **0.285** | **0.323** |
| Weather | 96 | 0.170 | 0.217 | **0.166** | **0.209** | 0.165 | 0.210 | **0.152** | **0.193** | 0.174 | 0.214 | **0.168** | **0.205** | 0.169 | 0.219 | **0.165** | **0.211** |
| | 192 | 0.216 | 0.257 | **0.211** | **0.252** | 0.214 | 0.252 | **0.204** | **0.241** | 0.221 | 0.254 | **0.219** | **0.252** | 0.225 | 0.265 | **0.222** | **0.260** |
| | 336 | 0.268 | 0.299 | **0.267** | **0.294** | 0.274 | 0.297 | **0.264** | **0.287** | 0.278 | 0.296 | **0.276** | **0.294** | 0.281 | 0.304 | **0.277** | **0.297** |
| | 720 | 0.340 | 0.351 | **0.337** | **0.346** | 0.350 | 0.345 | **0.344** | **0.339** | 0.358 | **0.347** | 0.356 | 0.348 | **0.359** | **0.354** | 0.362 | 0.355 |
| | Avg | 0.248 | 0.281 | **0.245** | **0.275** | 0.251 | 0.276 | **0.241** | **0.265** | 0.258 | 0.278 | **0.255** | **0.275** | 0.259 | 0.286 | **0.257** | **0.281** |
| Exchange | 96 | 0.114 | 0.242 | **0.104** | **0.231** | 0.086 | 0.207 | **0.083** | **0.202** | 0.086 | 0.206 | **0.085** | **0.205** | **0.105** | **0.235** | 0.109 | 0.240 |
| | 192 | 0.209 | 0.331 | **0.200** | **0.323** | 0.182 | 0.304 | **0.179** | **0.300** | **0.177** | **0.299** | 0.177 | 0.300 | 0.223 | 0.344 | **0.215** | **0.333** |
| | 336 | 0.379 | 0.452 | **0.370** | **0.446** | 0.332 | 0.418 | **0.317** | **0.408** | 0.331 | 0.417 | **0.330** | **0.416** | **0.363** | **0.439** | 0.366 | 0.439 |
| | 720 | 1.080 | 0.802 | **0.992** | **0.764** | 0.867 | 0.703 | **0.821** | **0.681** | 0.847 | 0.691 | **0.818** | **0.681** | 0.940 | 0.739 | **0.921** | **0.725** |
| | Avg | 0.446 | 0.457 | **0.416** | **0.441** | 0.367 | 0.408 | **0.350** | **0.398** | 0.360 | 0.403 | **0.353** | **0.401** | 0.408 | 0.439 | **0.403** | **0.434** |
| ILI | 24 | **1.870** | 0.913 | 1.962 | **0.909** | 2.103 | 0.972 | **2.007** | **0.932** | **2.438** | **1.076** | 2.450 | 1.082 | 1.806 | 0.893 | **1.715** | **0.857** |
| | 36 | **1.825** | **0.852** | 1.827 | 0.857 | 1.832 | 0.921 | **1.703** | **0.759** | 2.455 | 1.086 | **2.410** | **1.071** | 2.679 | 0.986 | **2.402** | **0.924** |
| | 48 | 1.824 | 0.834 | **1.764** | **0.827** | 2.224 | 0.998 | **1.877** | **0.725** | 2.580 | 1.118 | **2.513** | **1.095** | 2.584 | 0.938 | **2.224** | **0.843** |
| | 60 | 1.938 | 0.892 | **1.880** | **0.882** | 1.950 | 1.373 | **1.636** | **0.994** | 2.734 | 1.155 | **2.689** | **1.140** | 1.981 | 0.894 | **1.840** | **0.851** |
| | Avg | 1.864 | 0.873 | **1.858** | **0.869** | 2.027 | 1.066 | **1.806** | **0.853** | 2.552 | 1.109 | **2.516** | **1.097** | 2.263 | 0.928 | **2.045** | **0.869** |
| ECL | 96 | **0.144** | **0.244** | 0.149 | 0.248 | 0.139 | 0.235 | **0.138** | **0.233** | 0.148 | 0.240 | **0.145** | **0.234** | 0.168 | 0.272 | **0.165** | **0.266** |
| | 192 | **0.161** | **0.261** | 0.174 | 0.275 | 0.159 | 0.255 | **0.158** | **0.251** | 0.162 | 0.253 | **0.159** | **0.247** | 0.185 | 0.288 | **0.183** | **0.283** |
| | 336 | 0.227 | 0.328 | **0.197** | **0.293** | 0.176 | 0.272 | **0.173** | **0.270** | 0.178 | 0.269 | **0.173** | **0.262** | 0.204 | 0.306 | **0.193** | **0.293** |
| | 720 | 0.320 | 0.397 | **0.280** | **0.370** | 0.204 | 0.298 | **0.199** | **0.293** | 0.225 | 0.317 | **0.200** | **0.287** | 0.219 | 0.318 | **0.210** | **0.309** |
| | Avg | 0.213 | 0.307 | **0.200** | **0.296** | 0.170 | 0.265 | **0.167** | **0.262** | 0.178 | 0.270 | **0.169** | **0.258** | 0.194 | 0.296 | **0.188** | **0.288** |
| Traffic | 96 | 0.717 | 0.379 | **0.705** | **0.373** | 0.382 | 0.261 | **0.381** | **0.250** | 0.395 | 0.268 | **0.388** | **0.254** | 0.589 | 0.315 | **0.573** | **0.306** |
| | 192 | 0.708 | 0.377 | **0.682** | **0.364** | 0.396 | 0.267 | **0.389** | **0.255** | 0.417 | 0.276 | **0.410** | **0.264** | **0.618** | 0.324 | 0.623 | **0.320** |
| | 336 | **0.655** | 0.351 | 0.668 | **0.351** | 0.417 | 0.276 | **0.417** | **0.266** | 0.433 | 0.283 | **0.426** | **0.271** | 0.632 | 0.336 | **0.626** | **0.323** |
| | 720 | **0.709** | **0.374** | 0.730 | 0.392 | 0.460 | 0.300 | **0.461** | **0.287** | 0.467 | 0.302 | **0.461** | **0.289** | 0.659 | 0.349 | **0.643** | **0.328** |
| | Avg | 0.697 | **0.370** | 0.696 | 0.370 | 0.414 | 0.276 | **0.412** | **0.265** | 0.428 | 0.282 | **0.421** | **0.270** | 0.625 | 0.331 | **0.616** | **0.319** |

## A.3 FULL RESULTS

The full experimental results are reported in Tables 4 and 5. Without introducing any architectural modifications, simply replacing the original time domain loss with OLMA consistently improves the forecasting performance across models. Specifically, to respect the original supervision schemes of the various methods, all models except WPMixer employed the MSE loss, while WPMixer used the SmoothL1 loss Murad et al. (2025). For the hyperparameters $(\alpha, \beta, \gamma)$ of OLMA, we assign equal weights (0.34, 0.33, 0.33) for all models. Specifically, for the ECL and Traffic datasets, to mitigate the impact of entropy increase caused by channel-wise Fourier transform, we set the hyperparameters to (0.1, 0.45, 0.45).

Table 5: Full results of forecasting errors of OLMA on mainstream model baselines (TimeMixer, TimeXer, DLinear and WPMixer) which supervised by time domain losses (TDL) on different datasets. Lower values indicate better performance. The best results are highlighted in bold.

| Dataset | Length | TimeMixer TDL MSE | MAE | OLMA MSE | MAE | TimeXer TDL MSE | MAE | OLMA MSE | MAE | DLinear TDL MSE | MAE | OLMA MSE | MAE | WPMixer TDL MSE | MAE | OLMA MSE | MAE |
|---|---|---|---|---|---|---|---|---|---|---|---|---|---|---|---|---|---|
| ETTh1 | 96 | 0.375 | 0.400 | **0.370** | **0.388** | 0.382 | 0.403 | **0.378** | **0.391** | 0.375 | 0.399 | **0.365** | **0.386** | 0.347 | 0.383 | **0.345** | **0.379** |
| | 192 | 0.429 | 0.421 | **0.417** | **0.419** | **0.429** | 0.435 | 0.430 | **0.423** | 0.405 | 0.416 | **0.402** | **0.408** | 0.381 | 0.408 | **0.378** | **0.404** |
| | 336 | 0.484 | 0.458 | **0.472** | **0.443** | 0.468 | 0.448 | 0.470 | **0.442** | 0.439 | 0.443 | **0.430** | **0.428** | 0.382 | 0.412 | **0.379** | **0.408** |
| | 720 | 0.498 | 0.482 | **0.481** | **0.466** | 0.469 | 0.461 | **0.465** | **0.459** | 0.472 | 0.490 | **0.454** | **0.474** | 0.405 | 0.432 | **0.401** | **0.431** |
| | Avg | 0.447 | 0.440 | **0.435** | **0.429** | 0.437 | 0.437 | **0.436** | **0.429** | 0.423 | 0.437 | **0.413** | **0.424** | 0.379 | 0.409 | **0.376** | **0.406** |
| ETTh2 | 96 | 0.289 | 0.341 | **0.286** | **0.335** | 0.286 | 0.338 | **0.279** | **0.330** | 0.289 | 0.353 | **0.284** | **0.346** | **0.253** | 0.328 | **0.253** | **0.326** |
| | 192 | 0.372 | 0.392 | **0.365** | **0.386** | **0.363** | 0.389 | 0.365 | **0.384** | 0.383 | 0.418 | **0.370** | **0.402** | **0.303** | 0.364 | 0.304 | **0.363** |
| | 336 | 0.417 | 0.431 | **0.407** | **0.420** | **0.414** | 0.423 | 0.414 | **0.419** | 0.448 | 0.465 | **0.448** | **0.461** | **0.305** | 0.371 | **0.305** | **0.370** |
| | 720 | 0.419 | 0.440 | **0.414** | **0.434** | 0.408 | 0.432 | **0.395** | **0.422** | 0.605 | 0.551 | **0.556** | **0.525** | 0.373 | 0.417 | **0.371** | **0.413** |
| | Avg | 0.374 | 0.401 | **0.368** | **0.394** | 0.368 | 0.396 | **0.363** | **0.389** | 0.431 | 0.447 | **0.415** | **0.434** | 0.309 | 0.370 | **0.308** | **0.368** |
| ETTm1 | 96 | 0.320 | 0.357 | **0.311** | **0.343** | 0.318 | 0.356 | **0.311** | **0.345** | 0.299 | 0.343 | **0.298** | **0.338** | **0.275** | 0.333 | **0.275** | **0.329** |
| | 192 | 0.361 | 0.381 | **0.357** | **0.371** | 0.362 | 0.383 | **0.357** | **0.371** | 0.335 | 0.365 | **0.334** | **0.359** | 0.319 | 0.362 | **0.311** | **0.352** |
| | 336 | 0.390 | 0.404 | **0.388** | **0.394** | 0.395 | 0.407 | **0.389** | **0.394** | **0.369** | 0.386 | **0.369** | **0.379** | 0.347 | 0.384 | **0.346** | **0.378** |
| | 720 | **0.454** | 0.441 | **0.454** | **0.430** | 0.452 | 0.441 | **0.451** | **0.431** | 0.425 | 0.421 | **0.423** | **0.413** | 0.403 | 0.414 | **0.399** | **0.413** |
| | Avg | 0.381 | 0.396 | **0.378** | **0.385** | 0.382 | 0.397 | **0.377** | **0.385** | 0.357 | 0.379 | **0.356** | **0.372** | 0.336 | 0.373 | **0.333** | **0.368** |
| ETTm2 | 96 | 0.175 | 0.258 | **0.171** | **0.251** | 0.171 | 0.256 | **0.168** | **0.249** | 0.167 | 0.260 | **0.164** | **0.253** | 0.159 | 0.246 | **0.157** | **0.243** |
| | 192 | 0.237 | 0.299 | **0.235** | **0.295** | 0.237 | 0.299 | **0.232** | **0.291** | **0.224** | 0.303 | **0.224** | **0.289** | **0.214** | 0.286 | **0.214** | **0.281** |
| | 336 | 0.298 | 0.340 | **0.294** | **0.335** | 0.296 | 0.338 | **0.290** | **0.328** | 0.281 | 0.342 | 0.282 | **0.339** | 0.266 | 0.322 | 0.267 | **0.319** |
| | 720 | 0.391 | **0.396** | **0.390** | **0.396** | **0.392** | 0.394 | 0.393 | **0.391** | 0.397 | 0.421 | **0.383** | **0.408** | **0.344** | 0.374 | **0.344** | **0.370** |
| | Avg | 0.275 | 0.323 | **0.273** | **0.319** | 0.274 | 0.322 | **0.271** | **0.315** | 0.267 | 0.332 | **0.263** | **0.322** | 0.246 | 0.307 | **0.246** | **0.303** |
| Weather | 96 | 0.163 | 0.209 | **0.158** | **0.198** | 0.157 | 0.205 | **0.155** | **0.198** | 0.176 | 0.237 | **0.172** | **0.221** | **0.141** | 0.188 | **0.140** | **0.186** |
| | 192 | 0.208 | 0.250 | **0.206** | **0.243** | 0.204 | 0.247 | **0.202** | **0.242** | 0.220 | 0.282 | **0.213** | **0.260** | **0.185** | 0.229 | **0.185** | 0.230 |
| | 336 | **0.251** | **0.287** | 0.261 | **0.285** | 0.261 | 0.290 | **0.259** | **0.285** | 0.265 | 0.319 | **0.257** | **0.298** | 0.236 | **0.271** | **0.235** | **0.271** |
| | 720 | **0.339** | 0.341 | 0.341 | **0.338** | 0.340 | 0.341 | **0.338** | **0.337** | 0.323 | 0.362 | **0.320** | **0.351** | 0.307 | 0.321 | **0.306** | 0.322 |
| | Avg | **0.240** | 0.272 | 0.242 | **0.266** | 0.241 | 0.271 | **0.239** | **0.266** | 0.246 | 0.300 | **0.241** | **0.283** | 0.217 | **0.252** | 0.217 | **0.252** |
| Exchange | 96 | 0.083 | 0.201 | **0.082** | **0.200** | 0.087 | 0.206 | **0.085** | **0.205** | 0.081 | 0.203 | **0.080** | **0.202** | 0.094 | 0.216 | **0.092** | **0.212** |
| | 192 | **0.177** | 0.299 | 0.177 | **0.299** | 0.176 | 0.298 | **0.175** | **0.297** | 0.157 | 0.293 | **0.156** | **0.288** | 0.184 | 0.306 | **0.183** | **0.305** |
| | 336 | 0.329 | 0.413 | **0.320** | **0.409** | 0.346 | 0.425 | **0.338** | **0.421** | 0.333 | 0.441 | 0.365 | 0.452 | **0.339** | 0.421 | 0.340 | **0.421** |
| | 720 | 0.817 | 0.678 | **0.787** | **0.663** | 0.879 | 0.707 | **0.799** | **0.670** | 0.897 | 0.725 | **0.657** | **0.634** | 0.831 | 0.682 | **0.753** | **0.644** |
| | Avg | 0.352 | 0.398 | **0.342** | **0.393** | 0.372 | 0.409 | **0.349** | **0.398** | 0.367 | 0.416 | **0.315** | **0.394** | 0.362 | 0.406 | **0.342** | **0.396** |
| ILI | 24 | **2.245** | 0.985 | **2.245** | **0.953** | **2.203** | 0.958 | 2.205 | 0.958 | 2.215 | 1.081 | **2.119** | **0.964** | **1.349** | **0.731** | 1.432 | 0.745 |
| | 36 | 1.962 | 0.930 | **1.610** | **0.785** | 2.099 | 0.928 | **2.088** | **0.924** | 1.963 | 0.963 | **2.051** | **0.966** | **1.462** | **0.764** | 1.599 | 0.791 |
| | 48 | 2.393 | 1.086 | **1.563** | **0.785** | 2.081 | 0.977 | **2.064** | **0.928** | 2.130 | 1.024 | **1.992** | **0.961** | 1.813 | 0.882 | **1.525** | **0.788** |
| | 60 | 1.753 | 0.908 | **1.539** | **0.790** | 2.190 | 0.980 | **2.140** | **0.966** | 2.368 | 1.096 | **2.035** | **0.989** | 1.712 | 0.889 | **1.586** | **0.809** |
| | Avg | 2.088 | 0.977 | **1.739** | **0.828** | 2.143 | 0.961 | **2.124** | **0.944** | 2.169 | 1.041 | **2.049** | **0.970** | 1.584 | 0.817 | **1.536** | **0.783** |
| ECL | 96 | **0.153** | 0.247 | 0.155 | **0.246** | **0.140** | 0.242 | **0.140** | **0.239** | **0.140** | 0.237 | **0.140** | **0.236** | **0.128** | 0.222 | **0.128** | **0.221** |
| | 192 | **0.166** | **0.256** | 0.167 | 0.257 | **0.157** | 0.256 | 0.158 | **0.255** | **0.153** | **0.249** | 0.154 | **0.249** | **0.145** | 0.237 | **0.145** | **0.237** |
| | 336 | 0.185 | 0.277 | **0.184** | **0.273** | 0.176 | 0.275 | **0.176** | **0.272** | 0.169 | 0.267 | 0.169 | 0.267 | 0.161 | 0.256 | **0.160** | **0.253** |
| | 720 | 0.225 | **0.310** | **0.224** | 0.312 | **0.211** | **0.306** | 0.215 | 0.306 | **0.203** | 0.301 | 0.205 | **0.300** | 0.196 | 0.287 | 0.197 | **0.287** |
| | Avg | **0.182** | 0.273 | 0.183 | **0.272** | **0.171** | 0.270 | 0.172 | **0.268** | 0.166 | 0.264 | 0.167 | **0.263** | **0.158** | 0.251 | **0.158** | **0.250** |
| Traffic | 96 | 0.482 | 0.315 | **0.462** | **0.301** | **0.428** | 0.271 | 0.429 | **0.258** | **0.410** | 0.282 | **0.410** | **0.282** | **0.354** | 0.246 | **0.354** | **0.244** |
| | 192 | 0.486 | 0.315 | **0.476** | **0.299** | **0.448** | 0.282 | 0.456 | **0.268** | 0.423 | 0.287 | **0.422** | **0.286** | 0.371 | 0.253 | **0.367** | **0.251** |
| | 336 | 0.503 | 0.332 | **0.499** | **0.306** | 0.473 | 0.289 | **0.472** | **0.282** | 0.436 | 0.296 | **0.435** | **0.293** | 0.387 | 0.267 | **0.383** | **0.265** |
| | 720 | **0.524** | **0.326** | 0.545 | 0.326 | 0.516 | 0.307 | **0.513** | **0.300** | 0.466 | 0.315 | **0.464** | **0.311** | 0.431 | 0.289 | **0.427** | **0.285** |
| | Avg | 0.499 | 0.322 | **0.496** | **0.308** | **0.466** | 0.287 | 0.468 | **0.277** | 0.434 | 0.295 | **0.433** | **0.293** | 0.386 | 0.264 | **0.383** | **0.261** |

Table 6: Comparison of forecasting errors between OLMA and the combined OLMA + MSE loss across different datasets. Lower values indicate better performance. The best results are highlighted in bold.

| Dataset | Loss | 96 | | 192 | | 336 | | 720 | | Average | Improvement |
|---------|------|-----|-----|-----|-----|-----|-----|-----|-----|---------|-------------|
| | | MSE | MAE | MSE | MAE | MSE | MAE | MSE | MAE | | |
| ETTh1 | OLMA+MSE | 0.367 | 0.388 | **0.402** | 0.409 | **0.429** | 0.429 | 0.455 | 0.475 | 0.419 | **0.3%** |
| | OLMA | **0.365** | **0.386** | **0.402** | **0.408** | **0.429** | **0.428** | **0.453** | **0.474** | **0.418** | |
| ETTm1 | OLMA+MSE | **0.297** | **0.338** | 0.332 | 0.360 | 0.380 | 0.370 | 0.420 | 0.413 | 0.364 | **0.2%** |
| | OLMA | **0.297** | **0.338** | **0.331** | **0.359** | **0.379** | **0.369** | **0.419** | **0.412** | **0.363** | |
| Weather | OLMA+MSE | **0.171** | 0.223 | 0.212 | 0.262 | 0.258 | 0.303 | 0.322 | 0.357 | 0.264 | **1.4%** |
| | OLMA | **0.171** | **0.217** | **0.211** | **0.257** | **0.256** | **0.296** | **0.320** | **0.351** | **0.260** | |

## A.4 OLMA OR OLMA + TIME DOMAIN SUPERVISION?

An interesting question arises, that is, can incorporating time domain supervision into OLMA lead to better forecasting performance? In other words, does the pure frequency domain supervision of OLMA already capture all the essential information in the time series, or would adding time domain supervision introduce redundant information? To explore this, we conduct experiments using the basic DLinear model. Table 6 compares the forecasting errors of OLMA alone and OLMA combined with time domain supervision. It is evident that OLMA alone achieves better performance in most cases. This suggests that temporal domain supervision does not provide additional useful information on top of OLMA and may even introduce noise that harms performance in certain scenarios.

## A.5 IMPACT OF CHANNEL AND TEMPORAL LOSSES ON FORECASTING PERFORMANCE

The impact of the balancing channel and temporal losses of OLMA on forecasting error is further evaluated on the ETTh1, ETTm1 and Weather datasets with forecasting lengths of {96, 192, 336, 720}. Figure 5 illustrates the forecasting errors of WPMixer in different loss weight configurations. The results show that the performance of OLMA remains stable across a wide range of loss weight ratios between channel and temporal dimensions. This demonstrates that OLMA is a parameter-insensitive loss function, which can be seamlessly applied to any supervised method without the need for complex hyperparameter tuning.

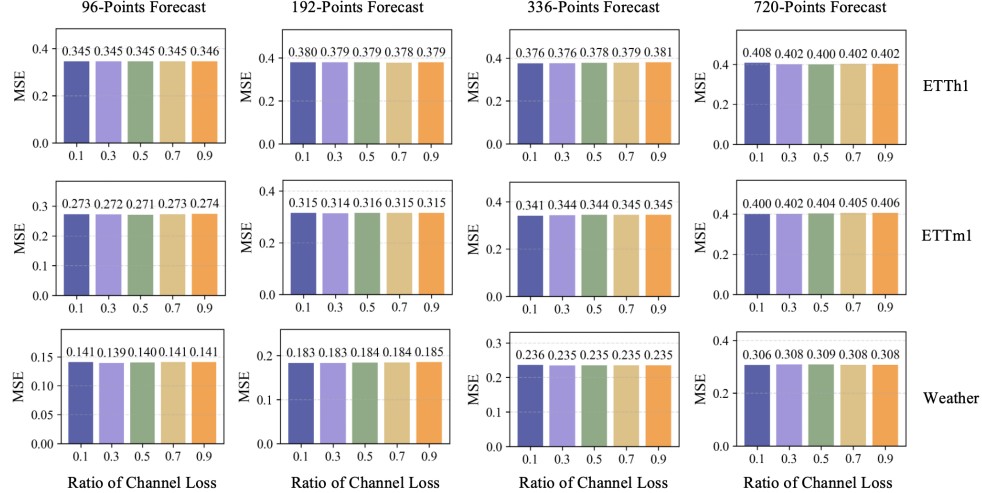

Figure 5: Impact of the ratio between channel and temporal dimension losses in OLMA on forecasting error on ETTh1, ETTm1 and Weather datasets under various forecasting lengths {96, 192, 336, 720} by WPMixer.

## B  THE USE OF LARGE LANGUAGE MODELS

We express our special thanks to the large language models GPT-4 and DeepSeek-R1 for their assistance in polishing writing.

