# OpenReview forum: "OLMA: One Loss for More Accurate Time Series Forecasting"
_ICLR.cc/2026/Conference — Submitted to ICLR 2026_

### Official Review · Reviewer_2rrV · 2025-10-21

**Soundness:** 3
**Presentation:** 3
**Contribution:** 2
**Rating:** 4
**Confidence:** 3

**Summary:**

This paper introduces OLMA, a theoretically grounded loss function for time series forecasting that integrates frequency-domain supervision to reduce entropy and mitigate frequency bias. The work is well-motivated, mathematically grounded, broadly applicable to a wide range of models, and clearly empirically validated on multiple benchmarks. However, its theoretical assumptions are somewhat idealistic, its distinction from previous frequency-domain methods such as FreDF is not clearly articulated, and its performance on high-dimensional datasets is weak. Overall, this paper presents a well-structured approach that combines information theory with practical loss design, but its novelty and generalizability require further verification.

**Strengths:**

1. This paper proposes a theoretical connection between prediction error and label entropy, providing a principled foundation for exploring time series forecasting.

2. The overall framework combines theory, loss design, and empirical validation in a coherent and easily interpretable manner.

3. The experimental results are comprehensive, spanning multiple datasets and various backbone models, and demonstrate consistent improvements.

**Weaknesses:**

1. This paper assumes that $\hat{x}$ is an unbiased estimator of $x$, but this property cannot be guaranteed in practice for complex neural network regressors or noisy time series labels.

2. Theorem 1 assumes that multiple Gaussian random processes are independent and identically distributed, which is difficult to prove for correlated multivariate time series. Real-world variables (for example, in ECL or traffic data) often have strong interdependencies.

3. Although OLMA introduces a loss-based formulation, earlier studies such as [1, 2, 3] have used frequency-domain error supervision. In addition to applying a DFT-based loss and providing theoretical support, this paper should more clearly explain its contribution.

4. Methods such as [4] have partially addressed the frequency-dependent learning bias issue through adaptive frequency fusion. The authors should clarify the contribution of OLMA relative to such methods.

5. The paper introduces several hyperparameters ($\alpha$, $\beta$, and $\gamma$), but does not explore how their values affect the results, nor does it investigate whether adaptive weighting can improve robustness across datasets of varying dimensionality.

[1] Xu, Zhijian, et al. “FITS: Modeling Time Series with 10k Parameters.” International Conference on Learning Representations (ICLR), 2024.

[2] Wang, et al. “FreDF: Learning to Forecast in the Frequency Domain.” International Conference on Learning Representations (ICLR), 2025.

[3] Yi, et al. “Frequency-domain MLPs are More Effective Learners in Time Series Forecasting.” Conference on Neural Information Processing Systems (NeurIPS), 2023.

[4] Zhang, et al. “Not All Frequencies Are Created Equal: Towards a Dynamic Fusion of Frequencies in Time-Series Forecasting.” ACM Multimedia (ACM MM), 2024.

**Questions:**

1. Why do the experimental results (Table 1) show that OLMA achieves significant improvements on low-dimensional datasets, but only modest or even negative improvements on multivariate datasets?

2. Theorem 1 assumes that multiple Gaussian random processes are independent and identically distributed but correlated across them. Can the authors clarify whether "independent and identically distributed'' holds? Does it refer to temporal independence within each process or independence between different variables? How can this condition be satisfied or approximated in real-world datasets?

3. What is the computational overhead introduced by OLMA during training, especially for high-dimensional inputs or long sequences? Does frequency-domain transformation significantly affect convergence?

---

### Official Review · Reviewer_D5QU · 2025-10-25

**Soundness:** 2
**Presentation:** 3
**Contribution:** 2
**Rating:** 4
**Confidence:** 4

**Summary:**

The paper proposes a new loss function, OLMA, for time series forecasting. It aims to solve two problems: inherent data noise and model frequency bias. OLMA applies a Discrete Fourier Transform (DFT) on the channel dimension, claiming this reduces data entropy. It also applies DFT and Discrete Wavelet Transform (DWT) on the temporal dimension. This provides direct supervision in the frequency domain to address model bias.

**Strengths:**

- The paper links channel correlations to data entropy and forecasting error. This is an interesting perspective.
- The proposed channel loss component $\mathcal{L}_{olma}^{(c)}$ is a novel idea.
- The experiments show that OLMA helps models learn low frequency components better (Figure 2).
- The method works on many models and datasets, showing good general use.

**Weaknesses:**

- The main theoretical claim is weak. Theorem 1 proves a unitary transform exists to reduce entropy. It does not prove that DFT is that transform. The paper seems to assume this without justification.
- The optimal transform for decorrelation is KLT (PCA), not DFT. Why was DFT chosen? This gap between theory and practice is a major problem.
- The paper's own results contradict its hypothesis. Figure 1 shows DFT increases entropy for the ECL dataset. But Table 1 shows OLMA improves performance on ECL.
- This ECL result suggests the performance gain might only come from the temporal loss

$\mathcal{L}_{olma}^{(t)}$.

It also suggests the channel loss

$\mathcal{L}_{olma}^{(c)}$

might be hurting performance, or working for reasons other than entropy.
- The theory (Section 3.1) discusses differential entropy of Gaussian noise $N$. The experiment (Section 4.1) measures Shannon entropy of the data $Y$. These are different measures on different variables. This makes the experimental validation (Fig 1) unconvincing.

**Questions:**

- Can you please justify using DFT for the channel loss? Theorem 1 suggests KLT (PCA) would be the optimal choice. Did you compare DFT to KLT?
- On the ECL dataset, channel DFT increases entropy, but OLMA improves results. Can you explain this contradiction? Does this mean the channel loss is not helpful for ECL?
- Your theory uses differential entropy on noise $N$. Your experiment uses Shannon entropy on data $Y$. Why is this a valid way to test your theory?

---

### Official Review · Reviewer_pF6H · 2025-10-27

**Soundness:** 3
**Presentation:** 2
**Contribution:** 2
**Rating:** 2
**Confidence:** 5

**Summary:**

This paper aims to (1) establish a theoretical forecasting error bound and (2) address frequency bias in time-series forecasting. The authors first prove that a unitary transformation exists which can reach the theoretical forecasting error bound, identifying the discrete Fourier transform (DFT) as one such solution. Building on this, they introduce the discrete wavelet transform (DWT) to further mitigate frequency bias. Finally, they propose a novel loss function, OLMA, and conduct a series of experiments to demonstrate the effectiveness of their approach.

**Strengths:**

1. The theoretical derivations supporting the proposed approach are interesting and appear solid.
2. The joint use of frequency and spatial domains within a unified framework is appealing for time-series forecasting tasks.
3. The topic of time-series forecasting is highly relevant to the ICLR community.

**Weaknesses:**

1. Recent studies have explored loss functions based on label transformation for time-series forecasting. In particular, FreDF [1] introduces a frequency-domain loss that applies the Discrete Fourier Transform (DFT) to both labels and predictions, minimizing their frequency-domain discrepancies. **The core formulation in the current paper (Eq.11), appears conceptually and mathematically identical to Eq.3 in [1].** However, **the paper does not acknowledge or discuss these prior contributions including [1], instead presenting the idea as novel.** This omission raises concerns of overstatement of originality and insufficient situating of the work within existing literature.
2. The authors state: `According to the maximum entropy theorem for continuous random variables with given mean and variance (Jaynes, 1957), for any random variable, its entropy is upper-bounded by that of a Gaussian with the same variance.` However, this theorem assumes the random variable is supported on $\mathbb{R}^{D}$, which may not hold in practice for many time-series datasets (e.g., electricity data is physically bounded). Please clarify the implications of this assumption and its potential impact on the theoretical results.

3. Why was the DFT specifically chosen as the unitary transformation? There are alternative approaches, such as invertible neural network architectures [2], which could also address the problem. What is the justification for preferring DFT over other possible unitary or invertible transforms?

4. The choice of the L1 norm is not justified in detail. What motivated this choice? Whether L1 norm outperforms the squared L2 norm (i.e., MSE) which in essence contribute to a significant fraction of performance improvement in this study? What would happen if other robust loss functions [3] were used instead in the transformed labels?

5. How are `high` and `low` frequency biases defined and what criteria are used to select the thresholds? Additionally, Figure 2 does not show error bars, making it difficult to assess the statistical significance of the results.

6. Given the symmetry property of the DFT for real-valued signals, **the frequency-domain representation should either be halved (length = 48 + 1 instead of 48, accounting for the zero-frequency terms) or explicitly symmetric (length = 96 + 1).** The paper should verify that this symmetry is correctly handled in the implementation and that the reported frequency-domain length (96) is consistent with the theoretical lengths of the FFT sequence.

7. The computational overhead of applying DFT and DWT transforms in the loss function is not reported. How does this compare to standard time-domain supervision, especially for long sequences?

8. In Appendix A.4, the authors conclude that OLMA alone outperforms OLMA + TIME DOMAIN. However, this argument is not entirely convincing because OLMA is a special case of OLMA + TIME DOMAIN where the time-domain loss weight is zero. Consequently, (i) the performance of OLMA can, in principle, be recovered within OLMA + TIME DOMAIN through appropriate hyperparameter tuning, and (ii) time-domain supervision is unlikely to be inherently detrimental, given its well-established effectiveness in related studies. Therefore, the observed result may stem from insufficient tuning or suboptimal weighting

9. The manuscript would benefit from substantial improvements in writing and organization. For example:
   - Figure 2 would be better placed in the `motivation analysis` section.
   - Lemmas and proofs should be moved to the appendix to improve the flow of the main text.

---
References:
[1]. FreDF: Learning to forecast in the frequency domain, ICLR'25
[2]. Transformed distribution matching for missing value imputation, ICML'23
[3]. Robust Regression, IEEE TPAMI
[4]. Reversible Instance Normalization for Accurate Time-Series Forecasting against Distribution Shift, ICLR'22.
[5]. Are transformers effective for time series forecasting?, AAAI'23
[6]. Autoformer: Decomposition transformers with auto-correlation for long-term series forecasting, NeurIPS'21

**Questions:**

Please see the abovementioned weaknesses.

---

### Meta-Review · Area_Chair_2B6v · 2026-01-01

**Summary:**

All reviewers have raised significant concerns regarding the paper’s core aspects and have unanimously recommended rejection. Furthermore, no author rebuttal was provided.

**Reviewer Scores:**

NA

---

### Decision · Program_Chairs · 2026-01-26

Reject